# Organic Matter in Riverbank Sediments and Fluvisols from the Flood Zones of Lower Vistula River

**Mirosław Kobierski** 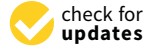 and **Magdalena Banach-Szott** * 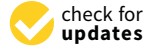

Department of Biogeochemistry and Soil Science, Bydgoszcz University of Science and Technology, 6 Bernardynska St., 85-029 Bydgoszcz, Poland; kobierski@pbs.edu.pl
* Correspondence: mbanach@pbs.edu.pl

**Abstract:** The research objective of this study was to determine whether and to what extent the form of use of Fluvisols (arable soil and grassland) of a Lower Vistula floodplain valley (Fordonska Valley, Poland) determined their relative organic matter properties, as compared with nearby riverbank sediments. Riverbank sediments were sampled from a depth of 0–20 cm, and soil samples from 0 to 30 cm, all in three replicates. Basic physico-chemical soil properties were determined: texture, pH, and the contents of total organic carbon (TOC), total nitrogen (TN), dissolved organic carbon (DOC) and dissolved organic nitrogen (DON). Humic acids (HAs) were extracted by the Schnitzer method and analysed to assess their spectrometric parameters in the UV–VIS range and hydrophilic and hydrophobic properties. Riverbank sediment samples contained significantly lower TOC and TN contents than Fluvisols, regardless of land-use type. The TOC, TN, DOC and DON contents and properties of humic acids in the Fluvisol surface layer depended on land-use type, because the arable soils had significantly lower TOC, TN, DOC and DON contents than the grasslands, despite having a similar grain size (texture). Based on the $A_{2/4}$, $A_{2/6}$, $A_{4/6}$ ratios, it was found that HA molecules isolated from the humus horizon of arable soils had a higher degree of maturity than HAs isolated from grassland soil samples. The spectrometric properties of humic acids isolated from riverbank sediments showed a higher degree of maturity than those from Fluvisols. This research showed that the properties of humic acids in Fluvisols are determined by the quantity and quality of organic matter transported in suspended matter that accumulates annually in flood valleys during flood events. The current land-use type of Fluvisols significantly influenced the properties of organic matter, and thus of humic acids. Therefore, these properties can be used to evaluate the transformation of organic matter that occurs in Fluvisols depending on the type of use.

**Keywords:** Fluvisols; riverbank sediments; humic acids; HPLC; UV–VIS

## 1. Introduction

The most extensive area of river floodplain Fluvisols in Poland is located along the Vistula River, which is characterized by a natural riverbank along its course; the channel is only stabilized in cities and at bridges. Floodplains with Fluvisols play a very important role, both agricultural and ecological, as some are landscape parks. The natural floodplain is delimited by a floodbank and comprises wasteland containing riparian woodlands and is under agricultural use as grasslands and arable land. Floodplain deposits reflect a diversity of mechanisms responsible for sediment transport and deposition, including transfer from the river channel during overbank flow, and as a result of slope wash from terraces and valley sides to distal floodplain parts [1,2]. Manuring practices are the main source of DOC and HS in this watershed where agricultural activity is predominantly focussed. The HS were then transferred to the river systems via runoff, particularly during the spring and autumn floods.

Geomorphological processes occurring in river floodplains affect the rate of development of fluvial landforms and usually involve an increase in the content of fine-grained

sediments rich in organic matter that accumulate after floodwaters recede [3–7]. Coarse-grained sediments are more heavily deposited in river channels or in their immediate vicinity. Fluvial sediments are deposited in the Lower Vistula floodplains every year, in spring and—less frequently—summer floods [8]. The depositional process can significantly enhance the nutrient pool of mineral sediments and accelerate initial ecosystem processes, including soil formation. The soil material in the zone adjacent to the channel is usually characterized by disturbed layering in the soil profile and a varied content of organic matter [9,10]. The quantity and quality of organic matter and grain-size composition may be key variables in the pedogenesis of alluvial soils (pedobiogeomorphological processes) [11]. Hoffmann et al. [12] have stressed a significant effect of the depositional environment and sedimentary facies on the content of total organic carbon (TOC), which increases with increasing clay content. Davies-Vollum and Smith [13] identified five factors that affect OM content in deposits on the modern avulsive floodplain: position relative to the avulsion belt, changes in local channel activity, distance from an active channel, presence of alluvial ridges, and developmental stage of avulsion. It is difficult to define unambiguously how soil properties change during inundation in periods of flooding, or to what extent they affect the quantity and quality of organic matter. Marie et al. [14] emphasize the importance of a manuring practice as the main source of DOC and humic substances in watershed. The HS were then transferred to the river systems via runoff, particularly during the spring and autumn floods.

Organic matter is a basic and key soil component [15–17], and plays an important role in, among other things, shaping the quality of the environment and global climate changes [18–21]. When introduced into the soil, the (quantitative) entirety of organic matter participates in the global cycle of environmental carbon. It is therefore a fundamental link in the sequestration of carbon and the release of $CO_2$ into the atmosphere [22].

Humic substances constitute a major component of organic matter [23,24]. They are formed by transformation and decomposition processes collectively known as "humification" [24–26]. The term "humic substances" is used to scientifically describe certain components with complex structures that can be isolated and fractionated in several ways [26]. Humin (H) is the insoluble fraction of humic substances; humic acids (HAs) are the fraction that is soluble under alkaline conditions and fulvic acids (FAs) are the fraction that is soluble under both alkaline and acidic conditions [27].

Humic substances have a number of specific chemical and physical properties, such as: a high sorption capacity [28–30] and high-energy absorption across the entire spectrum, especially in the UV and IR ranges of the electromagnetic spectrum [31–34].

The quality of organic matter is largely determined by the properties of humic acids, which constitute one of its main fractions and thus participate in all soil processes [35–37]. The tests, which, among other things, reflect the nature and origin of humic acids, include: $A_{280}$, $A_{465}$, $A_{665}$ UV–VIS range absorbance values of their solutions, and $A_{2/4}$, $A_{2/6}$, $A_{4/6}$, $\Delta logK$ coefficients of absorbance [33,38–42]. These can be used to determine the degree of advancement of the humification of organic materials, as well as changes in the properties of the humic acids that occur due to various anthropogenic factors.

The degree of humification of organic matter is also related to the hydrophilic–hydrophobic properties of humic acids. Woelki et al. [43], Preuße et al. [44] and Dębska et al. [45–47] obtained the separation of HA molecules into hydrophilic (HIL) and hydrophobic (HOB-1 and HOB-2) fractions. The relative proportions of the two fractions determine the solubility of humic acids and, as a result, their migration deep into the soil profile. According to reports in the literature [45], as the degree of humification increases, so too does the share of the hydrophilic fraction in humic acid molecules, while the hydrophobic fraction decreases. Consequently, HA molecules with a higher degree of "maturity" have a higher HIL/ΣHOB ratio.

Flood events are important because bottom sediments carried out of the river channel affect the physico-chemical and biological properties of floodplain soils, which have a relatively high content of humus [9,48,49]. The amount of soil material deposited depends

on the size and extent of the flood, and on the sediment load transported in the river. The Fluvisols that are formed can both indicate changes taking place within the river and allow changes going on in the entire catchment area to be interpreted, because their properties are the result of geological, pedogenic and anthropogenic factors [9]. The aim of this research was to evaluate the current properties of organic matter in samples of the surface layer of alluvial soils in floodplains annually inundated by floodwaters and in riverbank sediments of the nearby Vistula River channel. The overarching objective was to compare the properties of humic acids extracted from the riverbank sediments against those from the alluvial soils used as arable land and grasslands.

## 2. Materials and Methods

### 2.1. Materials

Samples of sediments and alluvial soils were collected from the Fordońska Valley within the Lower Vistula, protected area of Chełmiński Landscape Park (Figure 1). The Vistula is the longest and largest river in Poland, with a length of 1022 km. The Vistula basin covers 194,424 km$^2$. The Vistula has been in its present shape for over the last 14,000 years, after the complete recession of the Scandinavian ice sheet from the area. The research covered floodplain areas between the Vistula channel and a flood embankment 200 m to 600 m from the river channel.

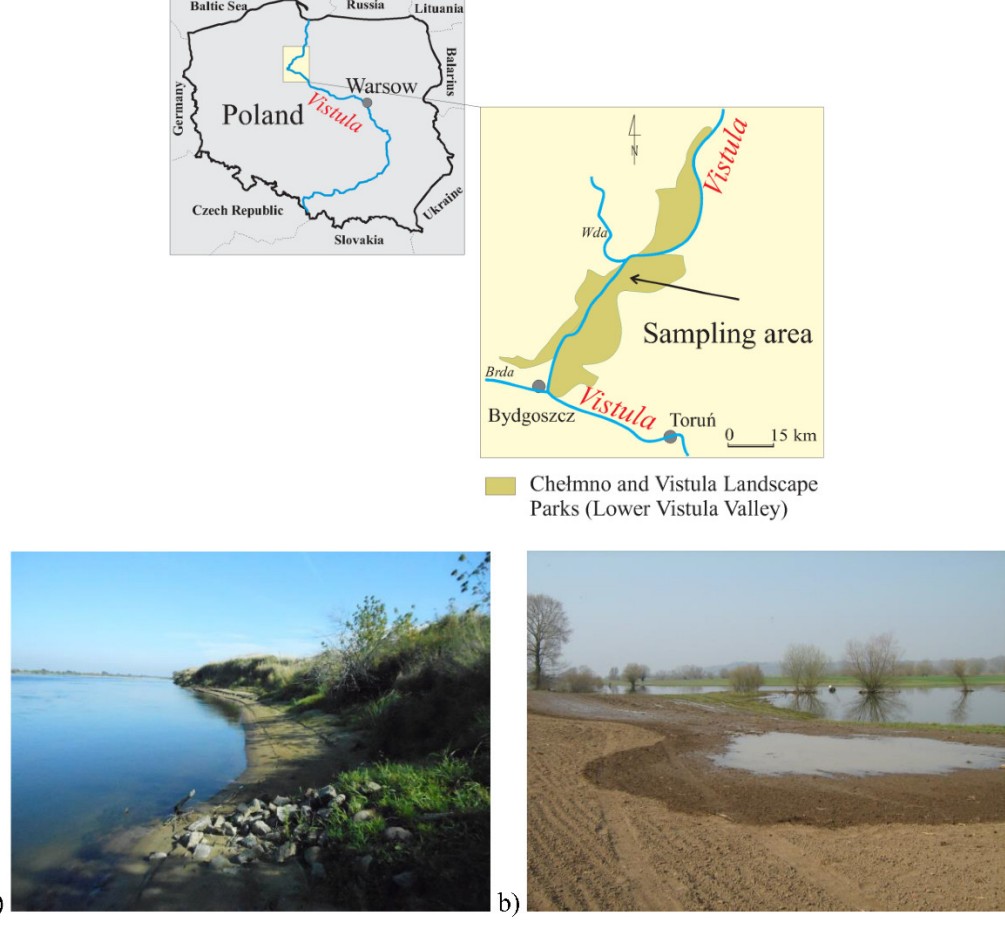

**Figure 1.** Schematic map and sampling places: (**a**) riverbank sediments; (**b**) floodplain.

Fluvisol samples were collected using a spade from a depth of 0–30 cm in three replicates: ten samples were taken from arable soils and eight from grassland. The distance between the replications was 20 m in an equilateral triangle and soil samples were taken in a distance within a few kilometres (floodplain). Soil samples were packed into plastic bags

and transported to a laboratory. Sediment samples were also collected in triplicate, from a depth of 0–20 cm. The sediments samples were similarly pre-treated. The samples were prepared in a laboratory for analysis as follows: plant residue removal, drying at room temperature, grinding, and sieving through a 2.0 mm mesh.

*2.2. Methods*

2.2.1. Basic Parameters of Soils

The following parameters were assayed:

- The contents of total organic carbon (TOC) and total nitrogen (TN).

The contents of total organic carbon, total nitrogen, dissolved organic carbon (DOC) and dissolved organic nitrogen (DON) were assayed with a Vario Max CN analyser provided by Elementar (Langenselbold, Germany).

The extraction of DOC and DON was performed with 0.004 M $CaCl_2$ for 1 h at a ratio of the soil to extraction solvent of 1:10 (*w/v*). The contents of TOC and TN are expressed in g $kg^{-1}$ of d.w. of soil [50]. The DOC and DON contents are expressed in mg $kg^{-1}$ of d.w. of soil sample [51].

- pH—in the suspension of 1 M KCl;
- The grain-size composition was determined applying the areometric method.

2.2.2. Extraction and Analyses of Humic Acids

Humic acids (HAs) were extracted and purified according to standard methods using the following procedure:

Decalcification (24 h) was achieved with 0.05 M HCl (1:10 *w/v*). After centrifugation, the residue was washed with distilled water until neutral.

Extraction (24 h) of the remaining solid was carried out with 0.5 M NaOH (1:10 *w/v*), with occasional mixing, followed by centrifugation.

Precipitation (24 h) of HAs was carried out from the resulting alkaline extract with 2 M HCl to pH = 2 and centrifugation.

The purification of the resulting HAs was carried out as follows: the humic acid residue was treated with a mixture of HCl/HF (950 mL $H_2O$, 5 mL HCl, 5 mL HF) over a 24 h period, followed by centrifugation. This procedure was performed three times. The HA residue was treated with distilled water until zero reaction with chloride was achieved.

The preparations were lyophilized and powdered in agate mortar. The ash content in the HA preparations was lower than 2%.

In the separated humic acids, the following determinations were made:

Hydrophilic and hydrophobic properties were determined with a liquid chromatograph HPLC Series 200 with a DAD detector from Perkin-Elmer, Shelton, USA. The separation involved the use of column X-Terra C18, 5 μm, 250 × 4.6 mm. The solutions of humic acids were applied in 0.01 mol/L NaOH of the concentration of 2 mg $L^{-1}$; injection of the sample—100 μL; solvent: acetonitrile–water; solvents flowed into the gradient (ratio of $H_2O$: ACN (*v/v*) over 0–6 min—99.5:0.5; 7–13 min—70:30; 13–20 min—10:90); and detection—at a wavelength of 254 nm. Based on the determined areas under the peaks, the share of hydrophilic (HIL) and hydrophobic (ΣHOB = HOB1 + HOB2) fractions in humic acid molecules and the parameter HIL/ΣHOB were determined [43–45].

UV–VIS absorption spectra (Perkin Elmer UV–VIS Spectrometer, Lambda 20, Ueberlingen, Germany): VIS spectra were obtained from 0.02% humic acid solutions in 0.1 M NaOH and UV spectra were obtained after fivefold dilution. The absorbances measured at 280 nm ($A_{280}$), 400 nm ($A_{400}$), 465 nm ($A_{465}$), 600 nm ($A_{600}$) and 665 nm ($A_{665}$) were used to calculate coefficient values:

$A_{2/4}$—280 nm and 465 nm absorbance ratio;
$A_{2/6}$—280 nm and 665 nm absorbance ratio;
$A_{4/6}$—465 nm and 665 nm absorbance ratio;
$\Delta logK = logA_{400} - logA_{600}$ (Kumada 1988).

Susceptibility to oxidation was determined with hydrogen peroxide through the measurement of decreased HA solutions' absorbance (0.02% HAs and 1.5% $H_2O_2$ in 0.1 M NaOH), at the wavelength of 465 nm (Perkin Elmer UV–VIS Spectrometer, Lambda 20, Ueberlingen, Germany). The susceptibility to oxidation was calculated from the following formula:

$$\Delta A_u 465 = \frac{A_0 - A_u}{A_0} \cdot 100\%$$

where: $A_0$—initial absorbance (prior to adding $H_2O_2$); $A_u$—absorbance after oxidation.

### 2.2.3. Statistical Analysis

The soil properties were determined with descriptive statistics: arithmetic mean, minimum value, maximum value, standard deviation, and coefficient of variation. The tables present the mean values from three replications. The statistical analyses of humic acids involved an analysis of variance. The significance of differences was evaluated using Tukey's test. Pearson's correlation was also analysed at a significance level of $p < 0.05$. The statistical analyses were performed using the Statistica MS 12.0 software (StatSoft Inc., Tulsa, OK, USA).

## 3. Results and Discussion

### 3.1. Basic Soil Parameters

The right bank of the Vistula regularly breaks in many places, and the amount of sediments deposited on the floodplain depends on the extent of the floodplain and local landscape conditions. The erosion processes observed in the river channel involve the loss of fine sediments at the footslope of the riverbanks during strong currents, especially spring floods. A greater amount of coarse sediments—gravel and sand—are transported by the river and deposited in the river channel. However, as a flood recedes, fine sediment (silt and clay fraction) transported along the river accumulates on the floodplain [8]. Therefore, the studied riverbank sediments differed significantly from the Fluvisols in the percentage of sand, silt, and clay fractions ($p = 0.0001$) (Table 1).

**Table 1.** The analysis of variance (ANOVA) with Tukey test.

| Parameter | Riverbank Sediments ($n = 10$) | Fluvisols ($n = 18$) | $p$ * | Arable Soils ($n = 10$) | Grassland Soils ($n = 8$) | $p$ * |
|---|---|---|---|---|---|---|
| TOC (g kg$^{-1}$) | 14.4 | 21.4 | 0.008 * | 16.1 | 28.1 | 0.0003 * |
| TN (g kg$^{-1}$) | 1.33 | 2.06 | 0.008 * | 1.6 | 26.3 | 0.004 * |
| TOC/TN | 11.5 | 10.5 | 0.12 | 10.2 | 10.9 | 0.218 |
| pH | 7.31 | 6.78 | 0.015 * | 6.86 | 6.67 | 0.52 |
| Sand (%) | 82.3 | 35.9 | 0.0001 * | 38.4 | 32.9 | 0.53 |
| Silt (%) | 12.0 | 44.9 | 0.0001 * | 43.9 | 46.1 | 0.74 |
| Clay (%) | 5.7 | 19.2 | 0.0001 * | 17.7 | 21.0 | 0.20 |
| DOC (g kg$^{-1}$) | 0.62 | 0.84 | 0.36 | 0.55 | 1.20 | 0.01 * |
| DON (mg kg$^{-1}$) | 44.8 | 54.2 | 0.48 | 44.9 | 65.7 | 0.038 |
| $A_{2/4}$ | 7.02 | 7.73 | 0.01 * | 7.36 | 8.19 | 0.0009 * |
| $A_{2/6}$ | 34.0 | 45.7 | 0.0004 * | 41.5 | 50.8 | 0.0005 * |
| $A_{4/6}$ | 4.82 | 5.88 | 0.0002 * | 5.63 | 6.20 | 0.0007 * |
| $\Delta A_u 465$ (%) | 57.3 | 54.0 | 0.122 | 51.2 | 57.5 | 0.002 * |

*—significance level; TOC—total organic carbon; TN—total nitrogen; DOC—dissolved organic carbon; DON—dissolved nitrogen.

The Fluvisols had significantly higher contents of silt and clay fractions. The results confirm previous studies of soils of the Lower Vistula floodplains [52–54]. Simansky [55] reports that differences in the chemistry and physical properties of Fluvisols have been significantly affected mainly by their use and soil management practices but not as a consequence of the flow gradient along the river.

The fluvial sediments transported by the river to the floodplain valley are rich in organic matter. The TOC content in riverbank sediments was variable, as indicated by the high coefficient of variation (CV = 73.4%) and averaged 14.4 g kg$^{-1}$ (Table 2). The highest mean TOC content was found in samples of soils used as grasslands, at 28.1 g kg$^{-1}$. A significant difference was observed in mean TOC content between arable lands and grasslands ($p$ = 0.0003) (Table 1). A similar trend was observed in nitrogen content. The average TOC/TN ratio in riverbank sediments was 11.5, while in the floodplain soils it was 10.5 (Table 1). Despite the significant differences in mean TOC and TN contents in arable lands and grasslands, the TOC/TN ratios were similar, amounting to 10.2 and 10.9, respectively.

**Table 2.** Values of pH and the organic matter properties.

| Parameters | pH (1 M KCl) | TOC (g kg$^{-1}$) | TN (g kg$^{-1}$) | TOC/TN | DOC (g kg$^{-1}$) | DON (mg kg$^{-1}$) |
|---|---|---|---|---|---|---|
| Riverbank sediments ($n$ = 10) | | | | | | |
| Min. | 7.05 | 5.7 | 0.2 | 9.5 | 0.19 | 13.5 |
| Max. | 7.43 | 37.2 | 3.9 | 14.3 | 1.68 | 121.0 |
| Mean | 7.31 | 14.4 | 1.3 | 11.5 | 0.62 | 44.8 |
| SD | 0.14 | 10.57 | 1.07 | 1.87 | 0.54 | 39.81 |
| CV (%) | 1.9 | 73.4 | 82.3 | 16.3 | 87.1 | 88.9 |
| Arable soils ($n$ = 10) | | | | | | |
| Min. | 5.52 | 12.2 | 1.2 | 9.1 | 0.34 | 33.6 |
| Max. | 7.39 | 20.1 | 2.1 | 12.4 | 0.82 | 58.0 |
| Mean | 6.86 | 16.1 | 1.6 | 10.2 | 0.55 | 44.9 |
| SD | 0.56 | 3.00 | 0.35 | 0.99 | 0.17 | 8.41 |
| CV (%) | 8.2 | 16.6 | 21.9 | 9.7 | 30.9 | 18.7 |
| Grassland soils ($n$ = 8) | | | | | | |
| Min. | 5.71 | 21.9 | 1.9 | 9.50 | 0.75 | 43.7 |
| Max. | 7.20 | 41.9 | 4.4 | 12.7 | 2.64 | 118.8 |
| Mean | 6.67 | 28.1 | 2.64 | 10.9 | 1.20 | 65.7 |
| SD | 0.57 | 6.45 | 0.85 | 1.17 | 0.65 | 26.20 |
| CV (%) | 8.5 | 22.9 | 32.2 | 10.7 | 54.2 | 39.9 |

SD—standard deviation; CV—coefficient of variation.

The pH values for riverbed sediments ranged from 7.05 to 7.43 and varied inconsiderably, as confirmed by the very low coefficient of variation (CV) (Table 2). Arable land and grasslands had similar pH values ($p$ = 0.52) (Table 1).

It should be noted that the soil organic matter contains initial materials at different stages of decomposition and large-molecule compounds of specific properties. From an ecological point of view, the fraction of organic matter soluble in water or in salt solutions of pH = 7 (DOM) is of great importance.

The mean DOC content was 0.62 g kg$^{-1}$ in riverbank sediments with a high coefficient of variation of 87.1%, 0.55 g kg$^{-1}$ with CV = 30.9% in arable soils, and 1.20 g kg$^{-1}$ with CV = 54.2% in grasslands (Table 2). The soils of the floodplain valleys varied significantly in the content of this fraction of organic carbon ($p$ = 0.01) (Table 1). The fact that the average

DOC content was the lowest in arable soils may result from the cultivation treatments conducted. According to Anger et al. [56] and Smreczak and Ukalska-Jaruga [57], in agricultural soils, the DOC content depends mainly on the type of organic material introduced into the soil. Agricultural practices can often reduce the DOC content in soil. Similar results were presented by Banach-Szott et al. [53]. Higher mean DOC contents in the riverbank sediments compared with their contents in the surface layer of Fluvisols were noted.

The DOC content in the studied riverbank sediments and Fluvisols was positively correlated with TOC content (Table 3). According to Watanabe et al. [58], DOC concentrations are linearly correlated with HS concentrations, and are thus directly associated with soil-derived organic matter inputs. It is worth mentioning that in riverbank sediments a significantly positive correlation between the clay fraction and DOC content was noted (Table 3). The impact of flood events on HS distribution was also observed by Marie et al. [14].

**Table 3.** Pearson linear correlation coefficient values, $p < 0.05$, in riverbank sediments ($n = 10$).

| Parameters | DOC | $\Delta A_u 465$ | HIL |
|---|---|---|---|
| Riverbank sediments ($n = 10$) | | | |
| TOC | 0.97 * | - | - |
| DOC | - | - | - |
| Clay | 0.70 * | - | |
| Arable soils ($n = 10$) | | | |
| TOC | 0.95 * | - | - |
| DOC | - | - | - |
| Clay | - | - | 0.75 * |
| Grassland soils ($n = 8$) | | | |
| TOC | 0.96 * | - | 0.86 * |
| DOC | - | - | 0.78 * |
| Clay | - | −0.82 * | - |

*—values of significant correlation coefficient; TOC—total organic carbon; DOC—dissolved organic carbon.

Comparing DON contents between riverbank sediments and Fluvisols, the relationship was similar as for DOC content (Table 1). The average DON content in riverbank sediments was 44.8 mg kg$^{-1}$, with a high coefficient of variation of 88.9% (Table 2). The land-use type was determined by the amount of DON in the surface layer of the tested Fluvisols: arable soil—44.9 mg kg$^{-1}$; grasslands—65.7 mg kg$^{-1}$ ($p = 0.038$) (Table 1).

The contents of TOC, TN, DOC and DON in the studied riverbank sediments were highly differentiated, as confirmed by the high coefficients of variation (Table 2).

The results confirm previous reports in the literature that the soil-use type significantly determines the share of the soluble fraction of organic matter [53,59].

*3.2. Hydrophilic and Hydrophobic Nature of Humic Acids*

Humic acid molecules are composed of both hydrophilic and hydrophobic structures. Using the HPLC method, HAs can be separated into hydrophilic (HIL) and hydrophobic (HOB) fractions [43,44,53,60]. The HA molecules isolated from riverbank sediments and Fluvisols had a similar mean share of HIL fractions, at 46.1% and 46.8%, respectively (Figure 2). The share of HOB fractions was higher, ranging from 53.2% for the humic acids of grasslands and arable soils to 53.9% for the humic acids of riverbank sediments. The HIL/ΣHOB ratios are a consequence of changes in the shares of the discussed fractions. According to Dębska et al. [46] and Kobierski et al. [54], the HIL/ΣHOB parameter is related to the degree of humification of organic matter. The parameter value increases as the degree of maturity of HA molecules increases. HA molecules isolated from riverbank sediments

and Fluvisols had similar hydrophilic–hydrophobic properties (at $p = 0.329$) (Figure 3). Despite the lack of significant differences between the values of the HIL/ΣHOB ratio, HA molecules isolated from riverbank sediments had the lowest values of this parameter (0.860), and the HAs of grasslands the highest values (0.894) (Figure 3). Lower values of the HIL/ΣHOB ratio indicate the presence of HA molecules characterized by a higher degree of aliphaticity, lower molecular weight and the presence of simple aromatic structures [46]. Both the share of HIL and HOB fractions and the value of the HIL/ΣHOB ratio indicate that humic acids isolated from riverbank sediments showed a higher content of humic substances in an advanced stage of decomposition than the arable land and grassland HA molecules. It is also worth emphasizing that the HIL/ΣHOB values obtained for humic acids in arable soils and grasslands by Banach-Szott et al. [53] were lower, ranging from 0.75 to 0.77. This means that the tested HA molecules had a slightly higher degree of maturity. The relative proportions of the two fractions determine the solubility of humic acids and, as a result, their migration deep into the soil profile. This was the most visible for the HA molecules isolated from grasslands, for which the share of the HIL fraction was found to correlate positively with the contents of DOC and TOC (Table 3). In turn, the humic acids isolated from arable soils were characterized by a positive correlation between the share of the HIL fraction and the content of the clay fraction (Table 3).

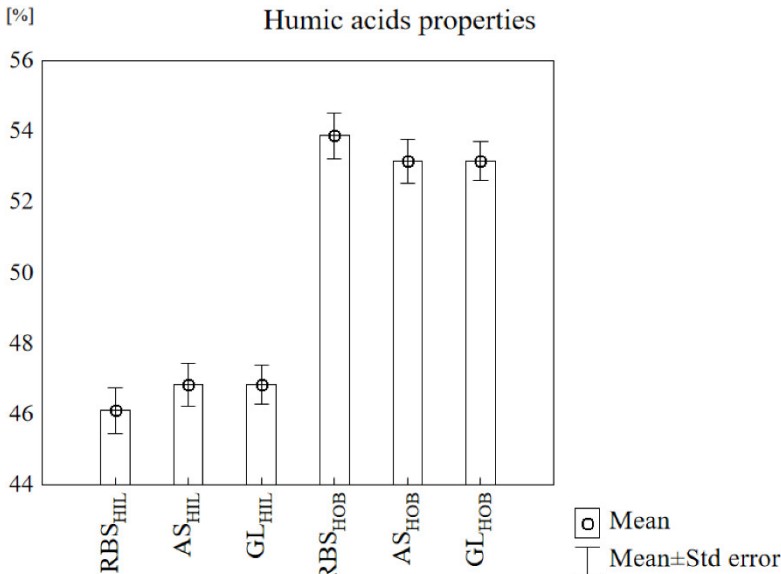

**Figure 2.** The percentage share of hydrophilic (HIL) and hydrophobic (HOB) fractions in humic acids isolated from riverbank sediments (RBS), arable soils (AS) and grassland soils (GL).

### 3.3. Spectrometric Parameters of Humic Acids in the UV–VIS Range

The HA molecules isolated from grasslands had the lowest intensity absorbance at wavelengths of 280 nm ($A_{280}$), 465 nm ($A_{465}$) and 665 nm ($A_{665}$), being 3.66, 0.448 and 0.072, respectively (Table 4). The highest absorbance value of 3.99 was at 280 nm, reflecting the content of lignin-type compounds, and was obtained for the humic acids in the arable soils. It is also worth noting that, for the HA molecules isolated from riverbank sediments, the highest degree of humification was obtained, as evidenced by the absorbance values at 665 nm, being 0.118.

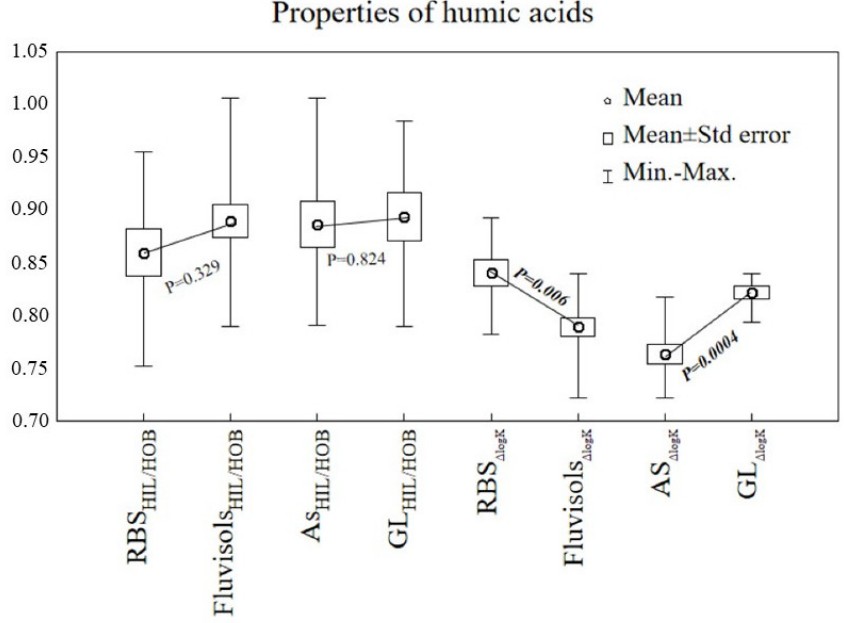

**Figure 3.** The hydrophilic–hydrophobic properties and parameters ΔlogK in humic acids isolated from riverbank sediments (RBS), arable soils (AS) and grassland soils (GL).

**Table 4.** Spectrometric properties of humic acids in the UV–VIS range of arable soils.

| Parameters | $A_{280}$ | $A_{400}$ | $A_{465}$ | $A_{600}$ | $A_{665}$ | $A_{2/4}$ | $A_{2/6}$ | $A_{4/6}$ | $\Delta A_u 465$ |
|---|---|---|---|---|---|---|---|---|---|
| Riverbank sediments (*n* = 10) | | | | | | | | | |
| Min. | 3.40 | 1.03 | 0.460 | 0.132 | 0.081 | 6.45 | 25.5 | 3.88 | 48.2 |
| Max. | 4.67 | 1.34 | 0.679 | 0.221 | 0.146 | 8.21 | 46.6 | 5.68 | 64.2 |
| Mean | 3.91 | 1.21 | 0.561 | 0.177 | 0.118 | 7.02 | 34.0 | 4.82 | 57.3 |
| SD | 0.36 | 0.11 | 0.07 | 0.03 | 0.02 | 0.55 | 6.09 | 0.59 | 4.56 |
| CV (%) | 9.2 | 9.1 | 12.5 | 16.9 | 16.9 | 7.8 | 17.9 | 12.2 | 7.9 |
| Arable soils (*n* = 10) | | | | | | | | | |
| Min. | 3.09 | 0.073 | 0.373 | 0.111 | 0.061 | 6.68 | 35.3 | 5.17 | 47.1 |
| Max. | 4.51 | 1.180 | 0.633 | 0.207 | 0.113 | 8.27 | 20.8 | 6.14 | 56.1 |
| Mean | 3.99 | 0.961 | 0.545 | 0.178 | 0.098 | 7.36 | 41.5 | 5.63 | 51.2 |
| SD | 0.41 | 0.318 | 0.070 | 0.028 | 0.015 | 0.492 | 4.95 | 0.31 | 2.93 |
| CV (%) | 10.3 | 33.1 | 12.8 | 15.7 | 15.3 | 6.7 | 11.9 | 5.5 | 5.7 |
| Grassland soils (*n* = 8) | | | | | | | | | |
| Min. | 3.05 | 0.706 | 0.350 | 0.103 | 0.056 | 7.98 | 47.0 | 5.84 | 52.5 |
| Max. | 3.94 | 0.970 | 0.489 | 0.152 | 0.081 | 8.73 | 54.8 | 6.59 | 64.8 |
| Mean | 3.66 | 0.893 | 0.448 | 0.135 | 0.072 | 8.19 | 50.8 | 6.20 | 57.5 |
| SD | 0.32 | 0.085 | 0.046 | 0.016 | 0.009 | 0.25 | 2.76 | 0.24 | 4.00 |
| CV (%) | 8.7 | 9.5 | 10.3 | 11.8 | 12.5 | 3.0 | 5.4 | 3.9 | 6.9 |

SD—standard deviation; CV—coefficient of variation.

As reported by Kumada [38], Tinoco et al. [61] and Filcheva et al. [62], lower values of absorbance and higher values of the $A_{2/4}$, $A_{2/6}$, $A_{4/6}$ and ΔlogK coefficients point to a chemical "young age" of HAs. Young humic acids show a lower degree of condensation of aromatic structures and a lower molecular weight, as compared with the HAs with a high

degree of humification. As shown in the data presented in Tables 1 and 4 and in Figure 3, the molecules of HAs differed in their degree of maturity.

The $A_{2/4}$, $A_{2/6}$, $A_{4/6}$ ratio values were significantly lower for the HAs isolated from riverbank sediments than for the Fluvisols' humic acids (Table 1). This evidences the highest level of humification among the HA molecules. It is worth noting that the humic acids of the grasslands had a lower degree of humification than did those of the arable soils—with significantly higher values of $A_{2/4}$, $A_{2/6}$, $A_{4/6}$ (Table 1). The obtained relationships between the values of the ratio $A_{4/6}$ from $A_{2/4}$ confirmed that the method of soil use determined the spectrometric properties of HAs (Figure 4).

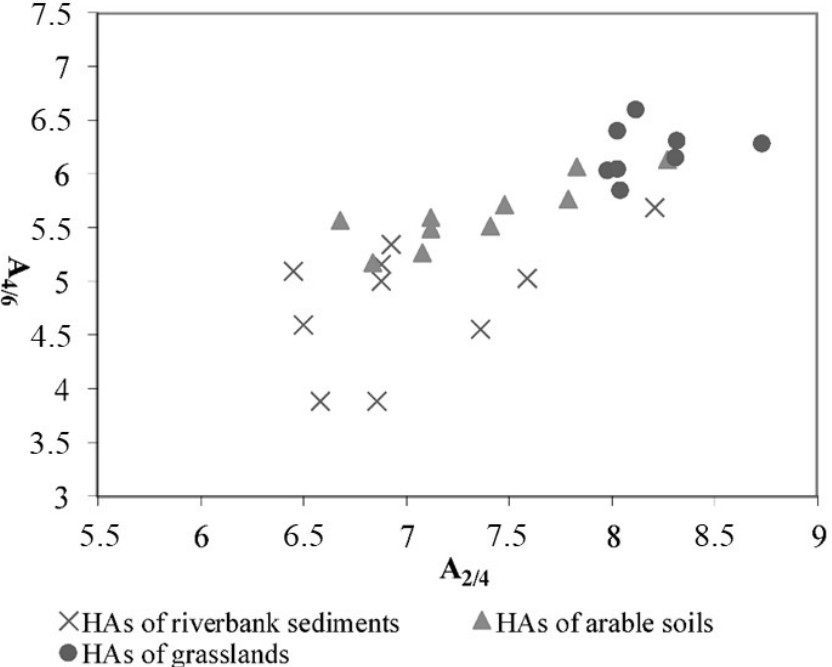

**Figure 4.** Dependency diagram of the values A4/6 on A2/4 ratio in the molecules of humic acids (HAs).

The higher $A_{4/6}$ ratio values obtained for the analysed HAs of grasslands reflect a lower content of humic substances at the initial decomposition stage as compared with the HAs of arable soils. Morán Vieyra et al. [41] indicated that the $A_{4/6}$ ratio value is higher for non-humified material due to the presence of proteins and carbohydrates. According to Albrecht et al. [63], a high $A_{4/6}$ ratio reflects a low degree of condensation of aromatic compounds. The $A_{4/6}$ ratio values for the HA molecules from the riverbank sediments were similar to those obtained by Polak et al. [42]. For humic acids isolated from sediments of 0.8 m depth collected from Goczałkowice Reservoir, the authors obtained a ratio in the range of 3.35 to 5.22. Meanwhile, the $A_{2/4}$, $A_{2/6}$, $A_{4/6}$ and $\Delta logK$ absorbance coefficient values for the humic acids isolated from the grasslands were much higher than those obtained by Drąg et al. [59] for HA molecules of meadow soil. Furthermore, Kobierski et al. [54], in a study of the humic acids of soils collected from a floodplain between the Vistula riverbed (Poland) and a flood embankment, obtained higher absorbance coefficient values than those for the HAs isolated from arable soils and riverbank sediments and tested herein. This confirms the high degree of transformation of the organic matter of the investigated arable soils and riverbank sediments and evidences the high quality of the humus. The results of the spectrometric analysis of the HAs were also consistent with the DOC content being higher in grasslands than in arable soils.

The susceptibility of humic acids to oxidation with the participation of $H_2O_2$ is measured by changes in the absorbance values of humic acids at a wavelength of 465 nm. It was shown that, as a result of reaction with $H_2O_2$, HA molecules had significantly lower

absorbance values at this 465 nm wavelength compared to the absorbance values obtained before the reaction.

HA molecules isolated from arable soils exhibited the lowest susceptibility to oxidation—with a mean of 51.2% (Table 1). Similar average values of susceptibility to oxidation, at the level of 57.3–57.5%, were obtained for humic acids extracted from grasslands and riverbank sediments (Table 1). According to Drąg et al. [59], the aliphatic part of humic acid molecules is more susceptible to oxidation, while the aromatic part is more resistant. This is confirmed by the significant differences between arable land and grassland humic acids ($p = 0.002$). The analysis of variance between the susceptibility to oxidation of HAs from riverbank sediments and those from Fluvisols did not confirm the relationship seen in the $A_{2/4}$, $A_{2/6}$, $A_{4/6}$ absorbance coefficients (Table 1).

The humic acids extracted from grasslands showed a significantly negative correlation between susceptibility to oxidation and the participation of clay (r = $-0.82$, $p = 0.05$) (Table 3).

Based on previous results of studies conducted depending on the distance from the riverbed [53], all Fluvisol samples for the current study were collected from a depth of between 200 and 600 m from the river channel, because the sampling zonality affected the properties of humic acids.

## 4. Conclusions

The riverbank sediment samples differed significantly from the Fluvisols in basic parameters such as TOC, TN and grain-size distribution. The properties of the Fluvisols of the Lower Vistula valley were determined by the land use the soils were subjected to. Arable soils had lower contents of TOC, TN, DOC and DON than did grassland soils.

The UV–VIS spectrometric properties of the humic acids isolated from riverbank sediments show them to be more mature than the humic acids isolated from Fluvisols.

The degree of maturity of humic acid molecules also depended on the type of land use. The humic acid molecules isolated from arable soils had a higher degree of humification than did the HA molecules isolated from grasslands.

**Author Contributions:** Conceptualization, M.B.-S. and M.K.; data curation, M.K. and M.B.-S.; formal analysis, M.K. and M.B.-S.; funding acquisition, M.K.; investigation, M.B.-S. and M.K.; methodology, M.K. and M.B.-S.; writing—original draft, M.K. and M.B.-S.; writing—review and editing, M.K. and M.B.-S. All authors reviewed the manuscript. All authors have read and agreed to the published version of the manuscript.

**Funding:** The research was made as part of 2716/B/P01/2011/40 research project, financed by the Ministry of Science and Higher Education (Poland).

**Institutional Review Board Statement:** Not applicable.

**Informed Consent Statement:** Not applicable.

**Data Availability Statement:** Data sharing not applicable.

**Conflicts of Interest:** The authors declare that they have no conflict of interest.

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
