# Peer review of "Organic Matter in Riverbank Sediments and Fluvisols from the Flood Zones of Lower Vistula River"

_agronomy, doi:10.3390/agronomy12020536_

Round 1

Reviewer 1 Report

Manuscript could be accepted after minor changes (see some comments in the text)

Author Response

Dear Reviewer,

The paper's Authors wish to thank for all the precious comments and guidelines. The manuscript has been checked by a native English speaker before submission.

We have made revisions according to your comments and suggestions, as described below.

I hope that the corrections introduced are satisfactory.

All the changes have been marked in the text.

With best regards,

Magdalena Banach-Szott

In the methodology part added methodology sampling and transportation and samples preparation site.

There was: Fluvisols samples were taken from a depth of 0–30 cm in three replicates: ten samples were taken from arable soils and eight from grassland. Sediment samples were also collected in triplicate, from a depth of 0–20 cm. The samples were prepared for analysis as follows: plant residue removal, drying at room temperature, grinding, and sieving through a 2.0 mm mesh.

There is: Fluvisols samples were collected using a spade from a depth of 0–30 cm in three replicates: ten samples were taken from arable soils and eight from grassland. The distance between the replications was 20 m in equilateral triangle and soil samples were taken in a distance within a few kilometers (floodplain). Soil samples were packed in plastic bags and transported to the laboratory. Sediment samples were also collected in triplicate, from a depth of 0–20 cm. The sediments samples were similarly pre-treated. The samples were prepared in laboratory for analysis as follows: plant residue removal, drying at room temperature, grinding, and sieving through a 2.0 mm mesh.

In the statistical analysis section, we have added information about the softwer used:

The statistical analyses were performed using the Statistica MS 12.0 software (StatSoft Inc., Tulsa, USA).

Reviewer 2 Report

Dear Editor,

Regarding the paper in question, I judge it has been written in a straightfoward way. The reading was easy.

My main concern is about the number of samples (very low). Please take into account it.

As minor suggestions, I would like to point out:

L16 and in the whole text: TN instead of Nt.

L36: "unregulated". It is confusing with law regulations. Please rephrase it.

L43: "As previous shown"? Is it correct here?

L58: Some authors [12]. Avoid this type. It would be better Hoffmann et al. [12].

L65: Some authors [14]: The same issue.

L130-135: The devices have been cited twice. Please rephrase.

L171: I think the equation is unformatted here (quite large typefont).

L184: Did you test the normality of the data? With the large CVs, it would be interesting use medians. Please consult a statician.

L281: Remove "e.g." here.

PS: Regarding plagiarism question, I don't feel qualified to judge it.

Sincerely yours. 

Author Response

Dear Reviewer,

The paper's Authors wish to thank for all the precious comments and guidelines.

We have made revisions according to your comments and suggestions, as described below.

I hope that the corrections introduced are satisfactory.

All the changes have been marked in the text.

With best regards,

Magdalena Banach-Szott

The number of samples:

soil and sediment samples were taken in triplicate and were representative for the studied region.

Line 16 and in the whole text: TN replaced throughout the text instead of Nt.

Line 36:

There was: The most extensive area of river floodplain Fluvisols in Poland is located along the Vistula River, which is characterized by an unregulated channel along its course; the river is only regulated in cities and at bridges.

There is: The most extensive area of river floodplain Fluvisols in Poland is located along the Vistula River, which is characterized by a natural riverbank along its course; the chanel is only stabilizated in cities and at bridges.

Line 43:

There was: As previously shown, manuring practice is the main source of DOC and HS in this watershed where agricultural activity is predominant.

There is: Manuring practices is the main source of DOC and HS in this watershed where agricultural activity is predominant.

Line 58:

There was: Some authors [12] have stressed a significant effect of the depositional environment and sedimentary facies on the content of total organic carbon (TOC), which increases with increasing clay content.

There is: Hoffmann et al. [12] have stressed a significant effect of the depositional environment and sedimentary facies on the content of total organic carbon (TOC), which increases with increasing clay content.

Line 65:

There was: Some authors [14] emphasize the importance of a manuring practice as the main source of DOC and humic substances in watershed.

There is: Marie et al. [14] emphasize the importance of a manuring practice as the main source of DOC and humic substances in watershed.

Line 130-135:

There was: The content of organic carbon and total nitrogen were assayed with a Vario Max CN analyser provided by Elementar (Germany). The content of TOC and Nt was expressed in g kgË—1 of d.w. of soil [50];

The content of DOC and DON were determined using a Vario Max CN analyzer produced by Elementar (Germany). The extraction was performed with 0.004 M CaCl2 for 1 h at a ratio of the soil to extraction solvent of 1:10 (w/v). The DOC and DON content was expressed in mg kgË—1 of d.w. of soil sample [51];

There is: The content of total organic carbon, total nitrogen, dissolved organic carbon (DOC) and dissolved organic nitogen (DON) were assayed with a Vario Max CN analyser provided by Elementar (Germany).

The extraction DOC and DON was performed with 0.004 M CaCl2 for 1 h at a ratio of the soil to extraction solvent of 1:10 (w/v). The content of TOC and TN was expressed in g kgË—1 of d.w. of soil [50]. The DOC and DON content was expressed in mg kgË—1 of d.w. of soil sample [51];

Line 171:

The formatting of the equation has been corrected (the font has been reduced).

Line 184:

Sample data has been drawn from a normally distributed population. The Student's t-test and the one-way  ANOVA were used for statistical analyses.

Line 281:

There was: As reported by, e.g., Kumada [38], Tinoco et al. [61] and Filcheva et al. [62],…

There is: As reported by Kumada [38], Tinoco et al. [61] and Filcheva et al. [62],…